# Wastewater plumes can act as non-physical barriers for migrating silver eel

Hendrik Volken Winter[1]*, Olvin Alior van Keeken[1], Frank Kleissen[2], Edwin Matheus Foekema[1,3]

**1** Wageningen Marine Research, IJmuiden, The Netherlands, **2** Deltares, Delft, The Netherlands, **3** Wageningen University, Marine Animal Ecology Group, Wageningen, The Netherlands

* erwin.winter@wur.nl

**Data Availability Statement:** All relevant data are within the paper and its Supporting Information files.

**Funding:** The author(s) received no specific funding for this work.

## Abstract

Non-physical barriers for migrating fish, such as effluent plumes discharged by sewage treatment plants (WWTPs), are hardly considered, and field studies on this topic are very scarce. The encounter with these plumes however may evoke behavioural responses in fish and could delay or (partially) block the migration. In this study, the behavioural responses of 40 acoustically-tagged silver eel (*Anguilla anguilla*) were monitored in situ, when confronting a WWTP effluent plume during their downstream migration in the canal Eems, the Netherlands. Their behavioural responses and the potential blocking effect of the plume were assessed using a 2D and 3D telemetry design displayed in the waterway, and matched to a modelled and calibrated WWTP effluent plume. When confronted with the WWTP effluent plume during their downstream migration, 22 of the silver eels (59%) showed an avoidance response, varying from lateral diverting to multiple turning in the vicinity of the effluent plume. Nineteen out of these 22 (86%) eventually passed the study site. No silver eel showed attraction to the plume. Delays in migration were from several hours up to several days. Due to the strong variation in discharged volumes and flow velocity of the receiving canal, the WWTP plume did not always flow over the full width of the canal. As a result, numerous migratory windows, where silver eels could pass the WWTP while avoiding direct contact with the plume, remained available in time. When discharge points cannot be avoided, reduced or restricted to areas that are not preferred as fish migration routes, discharge points should be designed such, that the chance is limited that a waterway is (temporarily) impacted over its full width.

## Introduction

Measures to improve the accessibility of rivers and waterways for migrating fish strongly focus on the removal or by-passing of physical barriers, such as dams, hydropower or pumping stations [e.g. 1–3]. Migrating fish can be hampered by inducing additional mortality when turbines are not designed in a fish-friendly manner [e.g. 4, 5], can show active avoidance or recurrence behaviour near or in front such man-made structures [6] resulting in additional energy loss and potential mismatch in arrival time on spawning grounds [7], or can be affected by blockage of their migration route [8].

**Competing interests:** The authors have declared that no competing interests exist.

There is less focus on measures to remove or alter non-physical barriers for fish, like sudden changes in water characteristics. When migrating along waterways near urban areas, fish may encounter effluents composed of complex chemical mixtures, that are discharged by sewage treatment plants (WWTPs). The encounter with such plumes may evoke behavioural responses, such as attraction or avoidance, which could lead to delays or even obstruction in migration. Downstream migrating fish have a higher probability to encounter a sharp change in water characteristics near a discharge point, compared to upstream migrating fish, which experience a more gradual change of water characteristics.

Next to the study on avoidance behaviour of Atlantic salmon (*Salmo salar)* to a spill of waste water discharge from a wood pulp factory [9], our study is the only other study to our knowledge that studies individual behavioural responses to a plume that is not related to oxygen, i.e. hypoxic conditions or super saturation. Furthermore, studies on the effect of thermal effluents on fish focus mainly on acclimatisation temperature and temperature preferences and not on individual behaviour [e.g. 10, 11]. Laboratory studies indicated that fish could detect a broad range of compounds in the water column, in some cases followed by behavioural responses [12–15]. A limited number of studies indeed indicated that avoidance behaviour occurred as a response to effluent from wood pulp industry [11, 16], plumes of water super saturated with oxygen from hydropower stations [17], or of hypoxic zones in catchment areas [18, 19]. The tendency of fish to avoid high carbon dioxide concentrations was used in aquaculture to steer fish [20], and the application of carbon dioxide to deter fish in a field situation was suggested [21, 22] and tested in situ in Canada, indicating that some species, such as common carp (*Cyprinus carpio*), avoided the $CO_2$ plume deployed [23].

In order to get more insight in the impact of such non-physical barriers, the behavioural responses of acoustically-tagged silver eel (*Anguilla anguilla*) were monitored in situ, when confronting an WWTP effluent plume during their downstream migration in the canal Eems, the Netherlands. This canal is an important shipping canal in the northern part of the Netherlands and is also important for fish migration, since it connects small brooks and other smaller canals to the Dutch part of the Wadden Sea. The goal of the study was to investigate if WWTP effluent discharges in this waterway impacted efficient downstream migration of silver eels.

2D and 3D acoustic telemetry was used to follow individual behaviour of migrating silver eels during two experimental periods. The first experimental period covered winter 2009 and spring 2010, the second was conducted during autumn 2010. The behaviours of the migrating silver eels were matched to the dynamics and spatial properties of an effluent plume, based on measurements and modelling. The behaviour of freely migrating silver eels were matched to the position of the plume they encountered.

## Materials and methods

### Study area

The study site was located in the Netherlands between the city Groningen and Delfzijl at the Eems canal, near the outlet of sewage treatment plant (WWTP) 'Garmerwolde', 53.24773 N, 6.67488 E (Fig 1). The 26 km stretch of the canal has an average water depth around 5.5 m and average width of 60 m. The shore line consists of riprap and steel sheet piles. The discharge point of WWTP Garmerwolde is situated 20 km upstream of the sluizes of Delfzijl, giving entry to the Dutch Wadden Sea.

### Telemetric equipment

For this study, a non-real-time underwater acoustic fine-scale positioning system was deployed, the VEMCO Positioning System (VPS). This system consists of VR2W receivers and

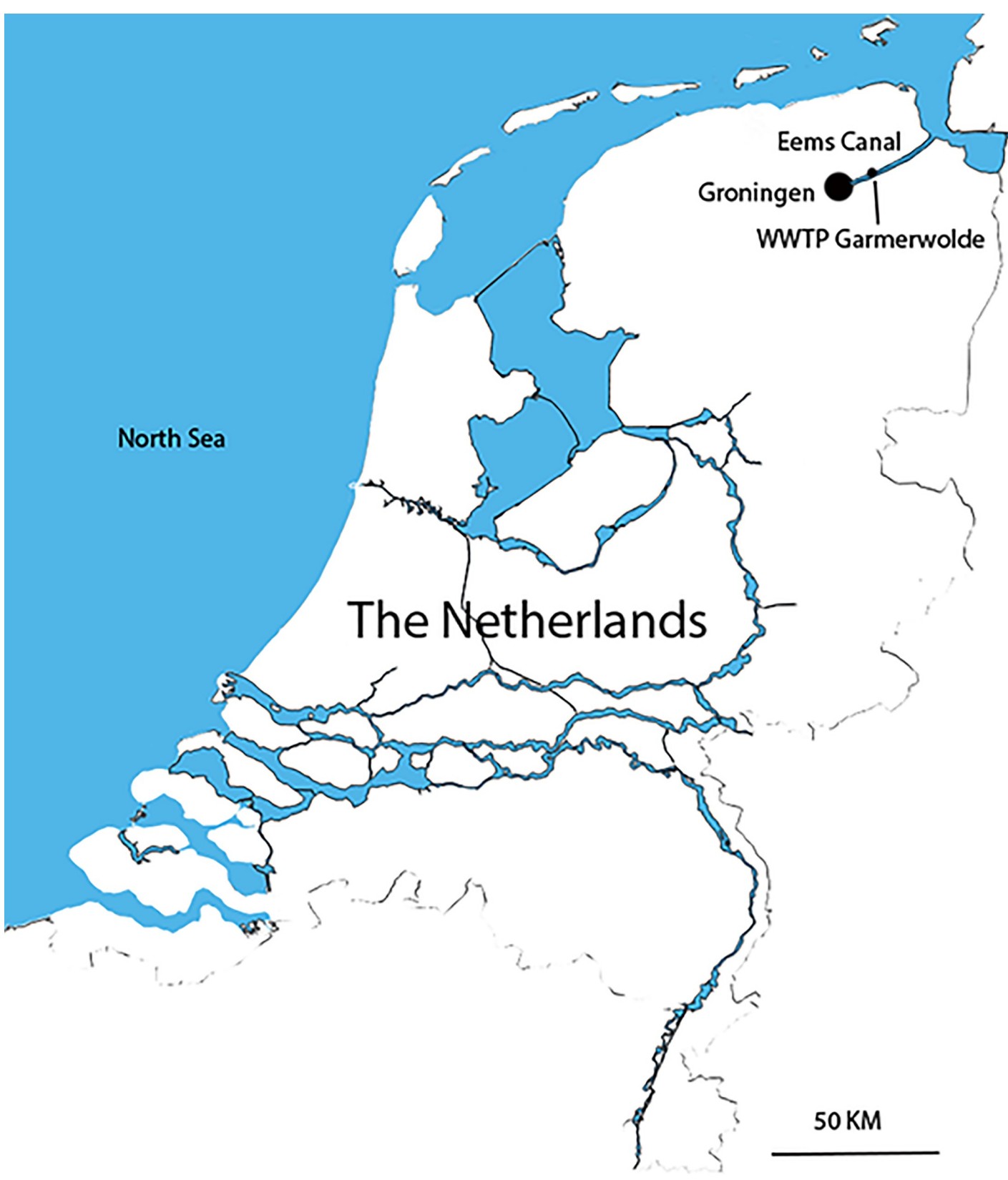

**Fig 1. The study area, the Eems canal in the northwestern part of the Netherlands.**

underwater acoustic transmitters (sync-tags), which were deployed in a grid of triangles or squares. With this system the position of a migrating fish could be calculated. Within the area covered by the triangles of receivers, the standard error cycle of the positioning had a diameter of 5 m, outside this area positioning is less accurate. Sync-tags, types V13 and V9T, were used with a random ping rate between 500 and 700 s. The use of the V9T temperature sync-tags is important for the calculation of the position of a fish, because large temperature differences between the effluent and receiving water could decrease the precision of a calculated position, as the speed of sound through water is temperature dependent [24].

The two field experiments were conducted from 4 December 2009 until 6 June 2010 (185 days), and from 27 October until 23 November 2010 (27 days). Although both experiments covered the year 2010, for readability they will be addressed as the 2009-experiment and the 2010-experiment in this manuscript, referring to the start year. At the field site, the receiver and transmitters were positioned underwater along the canal borders, moored along a line which was fixed at position with a weight at the bottom and a pop-up float. For the 2009-experiment, five receivers and four sync-tags were placed in the test area. For the 2010-experiment a total of 11 receivers and nine sync-tags were used, which allowed the test area to be expanded both upstream and downstream of the discharge point (Fig 2).

The silver eels were tagged with VEMCO coded acoustic V7-4L and V9-6L transmitters in the 2009-experiment, and V9P-6L transmitters in the 2010 experiment. All transmitters emit sound at 69-kHz. V7 and V9 transmitters were respectively 7 and 9 mm in diameter, 23 and 31 mm length, 1.0 and 2.9 g weight in water. The transmitters applied in 2010 (V9P-6L) also measured the water pressure with a converted resolution of 0.22 m, which enables determining the swimming depth of the tagged animal.

The receivers and sync-tags were retrieved after the study period and the data were processed by VEMCO to obtain calculated positions of the silver eels.

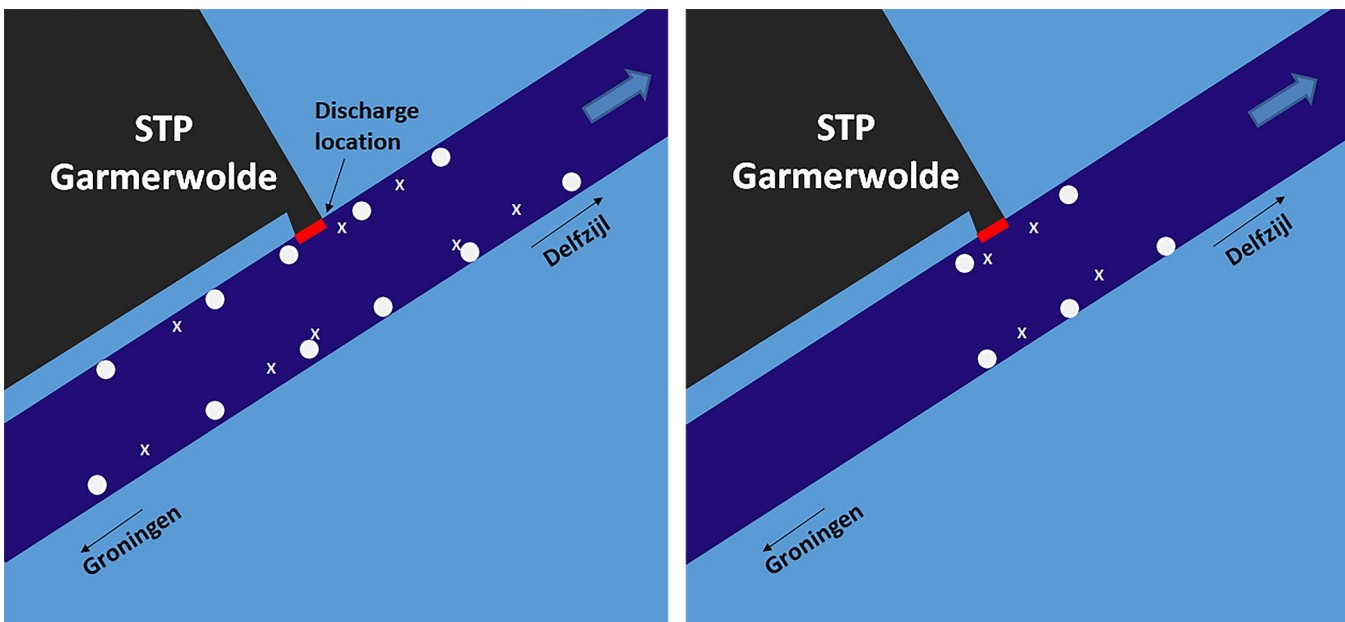

**Fig 2.** Vemco Positioning System (VPS) set-up at the test location in the 2009-experiment (left) with five receivers and four sync tags, and in the 2010-experiment (right) with 11 receivers and nine sync-tags. Receivers are indicated as white dots, sync-tags as white x. Blue arrow indicates the direction of the water flow which also is the direction for downstream migrating silver eel.

## Silver eel tagging

For each study period, 20 silver eels were used, caught with fykenets in the Eems canal approximately 20 km downstream of the study site. Fishing is prohibited in the upstream area of the canal. Before the start of the study, the silver eels were kept in aerated basins for one to five days. Morphometric features were measured in order to determine the sex and maturation stage according to Durif et al. [25]: total length (total length to the nearest cm), vertical and horizontal eye diameter (to the nearest 0.1 mm, only in 2010) and pectoral fin length (to the nearest 1 mm, only in 2010). The silver eels that were tagged showed visible signs of silvering, i.e. differentiated lateral line and white-silver ventral and black dorsal surfaces [26]. Lengths ranged from 57 to 93 cm for the first period and 77 to 102 cm for the second period. All silver eels were females, as males do not reach these sizes [27]. For the 2010 study, horizontal eye length ranged between 10 and 13.5 mm, vertical eye length between 8.6 and 12.2 mm and pectoral fin length between 37 and 46 mm.

The fish were anaesthetised in a 0.9 ml/l 2-phenoxy-ethanol solution. In each silver eel a 1–2 cm incision was made mid-ventral in the posterior quarter of the body cavity to implant a transmitter. The incision was then closed with a cyanoacrylate adhesive. After this surgical procedure that was performed according to Winter et al. [28] and Baras et al. [29], the silver eels were placed in an aerated recovery tank. Once they showed normal swimming behaviour, the silver eels were transported in an aerated water tank and released in the canal 2 km upstream from the effluent discharge point site.

The study protocol was approved by the committee on the ethics of animal experiments from Wageningen University and Research. Surgery was performed under anaesthesia, and all efforts were made to minimize suffering of the animals.

## Plume modeling

The 3D numerical model that was set up to describe the dynamic behaviour of the effluent plume, was based on Delft3D-FLOW [30] Delft3D-FLOW is a multidimensional (2D or 3D) hydrodynamic (and transport) simulation module within the Delft3D open Source Software (https://oss.deltares.nl/web/delft3d). Delft3D-FLOW calculates non-steady flow and transport phenomena that result from tidal and meteorological forcing on a rectilinear or a curvilinear, boundary fitted grid. The model can include features such as the effect of the earth rotation, space and time varying wind and atmospheric pressures, and simulation of thermal discharge. The main goal of the model was to describe the plume dynamic in time, using the variation in flow rate velocity, flow direction and water temperature in the canal in combination with the discharge volumes and temperature of the waste water plume.

The model for this study used a curvi-linear grid setup and covered a distance of approximately 5000 m along the Eems canal, with the south-western boundary at about 500 m upstream of the effluent discharge point and the north-eastern boundary at about 4500 m downstream. The grid resolution varied from about 3 m near the discharge point to 10 m at the longitudinal bounderies. In the vertical, the model had 10 layers, each with a thickness of 0.4 m, thus covering the local water depth of 4 m. The time step of the model was 15s.

Water level variation in the canal is in the order of centimetres, which is small compared to the total water depth and therefore neglected in the model. Currents in the canal, directed to the north-east, are induced by the outflow of excess water from the canal via sluices at Delfzijl Harbour 20 km downstream of the WWTP. Discharge only takes place during low tide in the Eems-Dollard estuary. When the sluices are closed during high tide, the canal may become stagnant. This can also occur during periods with low precipitation.

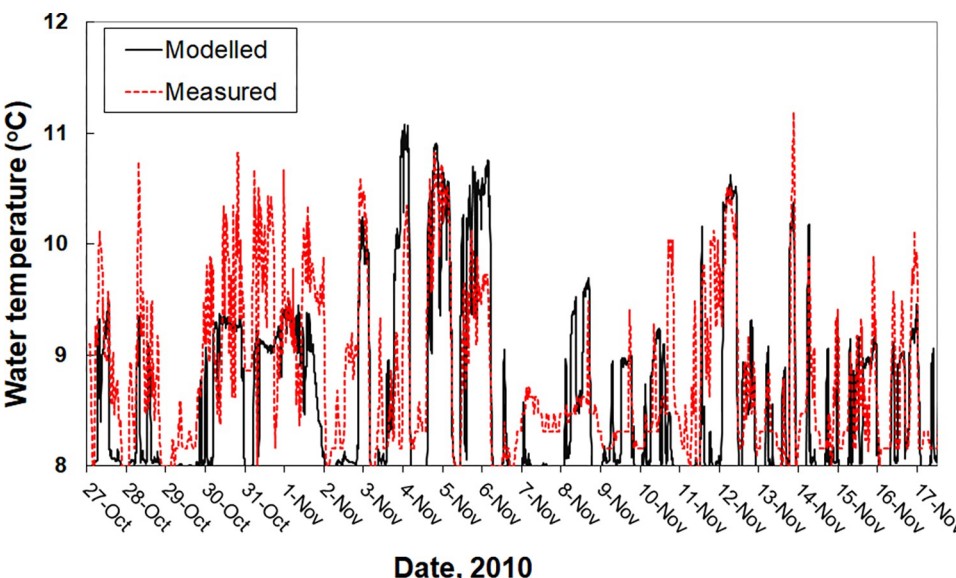

**Fig 3. Time line of observed and modelled temperature at a location in the Eems canal that is influenced by the discharge of the warmer sewage treatment plant (WWTP) effluent.**

Water flow velocity, used as model input, was measured during the test period in the canal upstream of the WWTP and was 0.2 ms$^{-1}$ maximum. Wind speed and direction were obtained from a nearby weather station (KNMI station 280, Eelde). Effluent discharge volumes were not recorded as such, but are directly correlated with the WWTP influent volumes that were provided by the operator of the WWTP (Waterboard Noorderzijlvest). Flow velocity of the effluent at the point of discharge was estimated to be 1 ms$^{-1}$. During the test period, the temperature of the effluent at the point of discharge was about 4°C higher than that of the water in the canal. Since temperature differences affect water density and hence the mixing processes, effluent and the canal water temperature was included in the model setup. In addition it allowed for an indicative validation of the model, with a reasonable reproduction of the dynamics of the observed water temperature in the area influenced by the effluent discharge (Fig 3).

The model outcome shows a dynamic behaviour of the effluent plume basically dictated by the water flow through the canal and the volume of the effluent being discharged. During stagnant periods the plume affects the whole canal area around the discharge point (Fig 4 left). In the presence of water currents, a long stretched plume along one side of the canal forms, leaving the opposite side unaffected (Fig 4 right).

## Data analysis

During the 2009-experiment, the limited number of five receivers and the use of standard transmitters without depth sensors allowed plotting tracks of individual silver eels in 2D only. Therefore, silver eel movements could only be matched with the approximate location of the upstream front of the plume. As more receivers and sync-tags were available for the 2010-experiment, the area in which the silver eels were tracked was almost doubled. Moreover, the transmitters used to tag the silver eels were equipped with depth sensors, allowing the tracking in a 3D field.

Positions of individual silver eels were calculated based on the VPS-position to create track lines. Plume dynamics were well captured within this time interval, but the telemetry data on

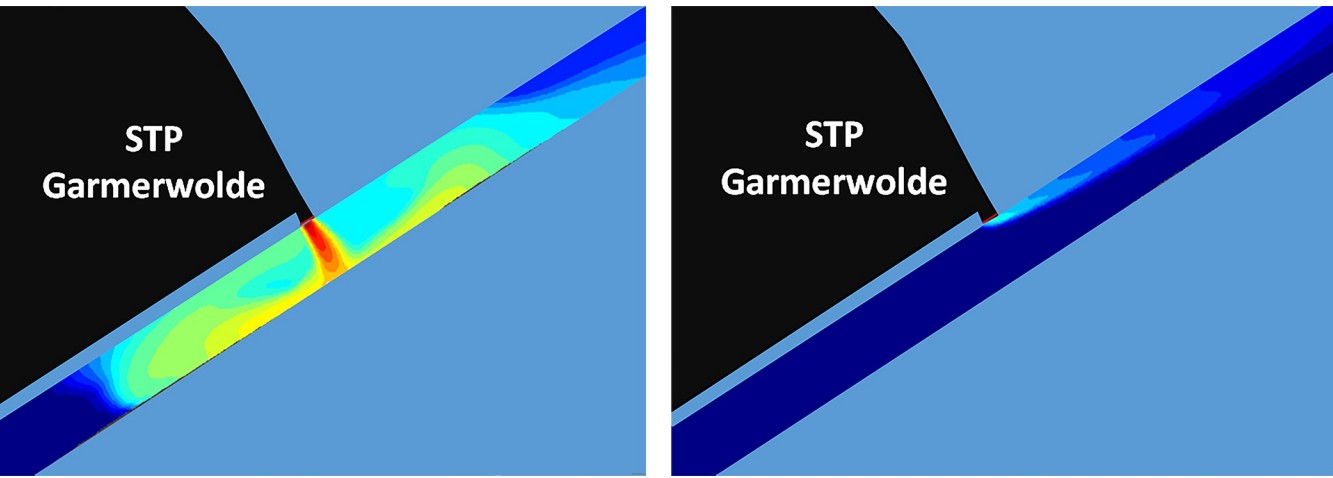

**Fig 4.** Example of plume mixing during a stagnant period (left) and in the presence of water currents (right) in the canal. The plume is indicated by temperature difference ranging from 0 (blue) to 0.6˚C (red).

silver eel movements only allowed making track lines with a time resolution with 20–35 s intervals between measured positions. To be able to link the silver eel movements to the behaviour of the plume, silver eel positions were interpolated to 15 s intervals from the available data points. As this interval was only slightly less than the interval of the data set, unsubstantiated precision in the track lines was avoided. The calculated positions of each individual silver eel was plotted as 2D track lines on top of the visualisation of the modelled effluent plume at the moment the silver eel entered the study area. In the modelling of the plume, temperature was used to identify plume influence, and when temperature increased >0.1˚C from upstream ambient canal temperature, silver eel were considered to be confronted with effluent influence.

Movement patterns of downstream migrating silver eels were categorized into different type of behaviours. Three main behavioural responses are thinkable when a silver eel is confronted with the effluent plume and different subcategories were defined based on the observed patterns (Fig 5):

1. No response to plume

   a. Passing through the study site, with no diversion from the plume

   b. Passing through the study site, showing random turns with no relation to the front or study site

   c. Passing through the study site, showing extensive random movements with no apparent relation to the plume front or study site

2. Avoidance behaviour

   a. No passage of the study site, return behaviour when confronted with plume,

   b. Passing through the study site, but avoidance of the plume during passage

   c. Passing through the study site, with multiple turns associated with meeting the plume

   d. Passing through the study site, but with extensive movements and turns, predominantly outside the plume area

3. Attraction behaviour

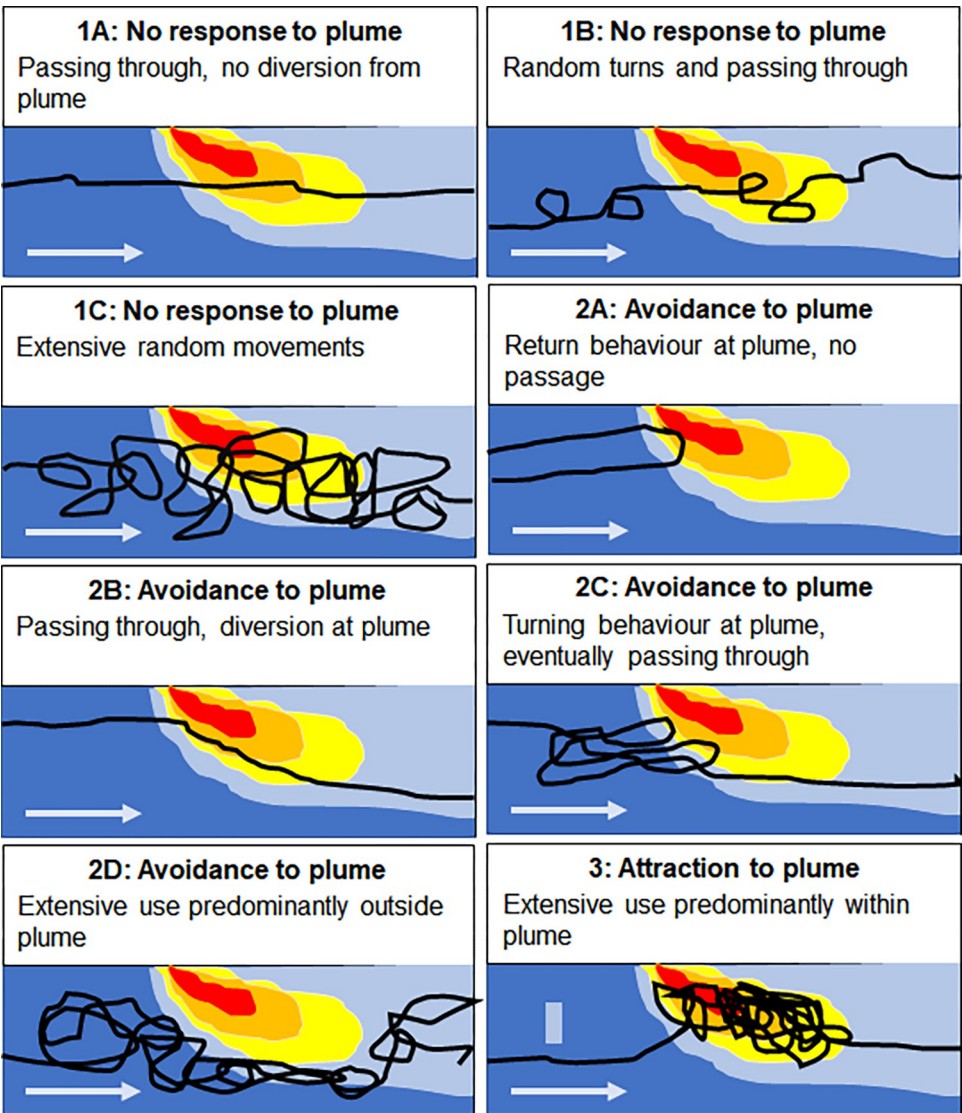

**Fig 5. Classification of the different silver eel behaviour subcategories.** Black line indicates fictional silver eel movement pattern as examples of the different potential behavioural response categories (silver eel entering site from the left along with flow direction (blue arrow)). The colours of the plume indicate plume strength, ranging from red for strong plume strength to blue with no influence.

 a. passing through, but staying in the plume area for a longer period. When meeting the plume, movement activity is centred in the area with a relative high effluent concentration.

## Results

In the 2009-experiment, 18 out of the 20 silver eels entered the study site at the effluent discharge point (Table 1). In the 2010-experiment this number was 19 out of 20. Individual variation in movement patterns was large, varying from fast directed movements to extensive backward and forward movements (Fig 6, see S1 and S2 Appendices). For the 2009-experiment, the interpretation of the responses was hampered by the limited surface of the study area. The responses of one individual fish (#3) could not be characterised.

**Table 1. Description of movement category for each silver eel.**

| Eel 2009 | Behavioural code | Short description | Eel 2010 | Behavioural code | Short description |
|---|---|---|---|---|---|
| 1 | 2B | Diversion at plume | 1 | 2D | Extensive movements outside plume |
| 2 | 2B | Diversion at plume | 2 | 1B | No response, random turning |
| 3 | - | Unclear | 3 | 1C | No response, random movements |
| 4 | 2A | Turning, no passage | 4 | 2C | Diversion at plume |
| 5 | 1B | No response, passing through | 5 | 2B | Turning, eventually passage |
| 6 | 1B | Turning, no passage | 6 | 2D | Extensive movements outside plume |
| 7 | 2B | Turning, eventually passage | 7 | 2A | Turning, no passage |
| 8 | 2C | Turning, eventually passage | 8 | 2C | Diversion at plume |
| 9 | 1A | No response, passing through | 9 | 2C | Diversion at plume |
| 10 | 1A | No response, passing through | 10 | 1C | No response, random movements |
| 11 | 2B | Diversion at plume | 11 | 1A | No response, passing through |
| 12 | 1A | No response, passing through | 12 | 1C | Extensive movements outside plume |
| 13 | - | Not detected | 13 | 2D | No response, random movements |
| 14 | 1A | No response, passing through | 14 | 1A | Diversion at plume |
| 15 | 1A | No response, passing through | 15 | 2C | Diversion at plume |
| 16 | - | Not detected | 16 | 2C | Diversion at plume |
| 17 | 2C | Diversion at plume | 17 | - | Not detected |
| 18 | 2B | Turning, eventually passage | 18 | 2D | Extensive movements outside plume |
| 19 | 2A | Turning, no passage | 19 | 1A | No response, passing through |
| 20 | 2B | Turning, eventually passage | 20 | 1B | No response, random turning |

In both experiments, the majority of the silver eels showed avoidance behavior when being confronted with the effluent plume (61% and 57% in the 2009 and 2010-experiment respectively, Fig 7). The other silver eels appeared to be indifferent to the plume. None of the silver eels showed attraction to the plume. There were some differences in subcategories between years within each type of response that partly is due to the larger study area in the 2010-experiment that allowed a longer observation of the movements of the individual fish. In the 2009-experiment some individuals (#1, 2, 5, 11, 15) entered and left the study area several times during the study period. The general results in both experiments were however very similar.

Of all the 22 silver eel that showed an avoidance response, 19 (86%) individuals eventually passed the study site. The remaining three silver eels turned when confronted with the plume and were not observed to pass the study area within the study period. The behavioural responses were most pronounced in the horizontal (2D) dimension, characterised by changing swimming direction or moving sideways away from the waste water outlet. In the 2010-experiments also depth profiles were available, showing that the detected silver eels swam at varying depth, frequently changing from surface layer to bottom water layer and dwelling on the bottom during more stationary periods. No obvious links were found with encountered effluent concentration that generally decreased with water depth. No indications were found that silver eels tried to avoid the plume by moving to deeper water layers. Delays in migration were from several hours up to several days.

## Discussion and conclusions

Of the 37 silver eels that were followed for this study, 59% showed an avoidance response when confronted with the effluent plume during downstream migration. In 14% of the cases, silver eels turned around after encountering the plume, left the study area in upstream

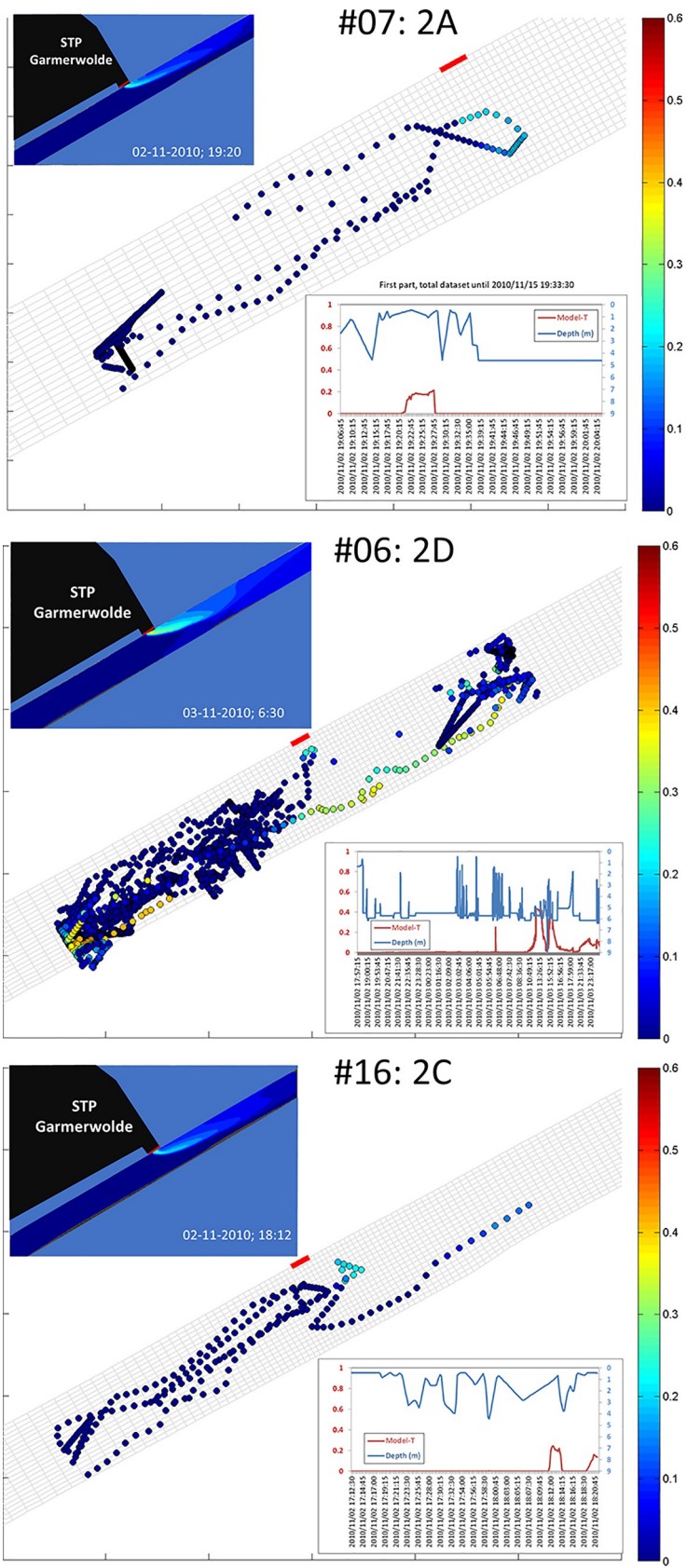

**Fig 6. Examples of subcategory movements of three downstream migrating silver eels within the test area of the 2010 (see S1 Appendix for 2009 and S2 Appendix for 2010 individual silver eel movements).** 2A: turning, no passage; 2D: extensive movements outside plume; 2C: diversion at plume. Left panel in each of the three examples: visualization of the effluent plume at moment of contact. Middle panel: horizontal movements of the eel within the test area, each dot represents one signal, colour indicates water temperature excess (proxy for contribution of effluent) as experienced by the fish. Right panel: vertical movement of eel (blue line) and modelled excess water temperature (red line).

direction and were never seen again during the study period. Whether these silver eels did or did not resume their migration after the experiment was ended, could not be determined from our data. Another explanation could be that these individuals were less motivated to migrate this year, as has been observed before among silver eel [28, 31].

The cues that triggered the observed behavioural responses were not the focus of this study and remain unclear. As silver eels are known for their keen smell [32] and an effluent plume contains a very diverse cocktail of components, we consider chemical cues as the most likely explanation for the behavioural responses that were observed. Since WWTP effluents are complex mixtures that can vary in time and location, the identification of the substances that trigger these responses will only be possible after a detailed study. Also temperature differences between the plume and adjacent water in the canal could have triggered a behavioural response, since silver eels can prefer certain water temperatures during migration [33]. Another potential cue could be water current [34]. Migrating silver eel encounter many differences in flow direction during their migration through river and canal systems. Silver eels that moved sideways next to the plume, might just followed the extra side component in flow. However returning in response to an increased side flow also occurred. According to Piper et al. [35], silver eels near a hydropower facility in the UK displayed erratic behaviour on encountering flow acceleration. The magnitude of the response was positively related to maximum water velocity and flow acceleration. Due to the strong variation in discharged volumes and flow velocity of the receiving canal, numerous migratory windows (moments silver eels could pass without being in contact with the plume) remained available in time and different water depths. Therefore we estimated that the chance that this type of effluent plumes completely blocked a migration path was small, although it may have caused delays in migration. However migrating silver eels can be subjected to a multitude of distracting and attracting flows, and a sequence of many effluent discharge points along the migration route might result in substantial migration delays or extra energy dissipation.

The results were very similar for both study periods with 56% and 63% of the silver eels showing avoidance behaviour in the 2009 and 2010-experiments respectively. Note that only one study location was covered. This location was selected because the canal is very monotonous in habitat quality and shore line, at least in the area not affected (upstream) by the

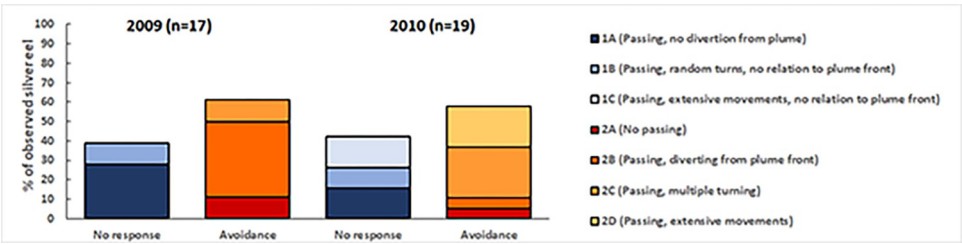

**Fig 7. Distribution of behavioural responses of silver eels when confronted with the effluent plume, during the 2009 (n = 18) and 2010 (n = 19) experiments.** For one eel in 2009 the movement was unclear.

WWTP effluent. This did not only help interpretation of the silver eel behaviour, but also allowed a reliable modelling of the dynamic behaviour of the effluent plume. Behavioural changes could reliably be linked to the presence of the effluent plume. Behavioural patterns, however, substantially differed between individuals. This might be related to differences in personality, i.e. behavioural differences among individuals that are consistent through time and across contexts, which were recently found to play a role in passage success of juvenile eels [36]. It might be interesting for future studies to perform personality assays prior to tagging silver eels to address this.

The results of this study cannot be translated directly to other locations, where effluent plumes will differ in dynamic behaviour, as this varies with discharge volumes and local conditions (e.g. currents, flow velocity, water depth and width) of the receiving surface water. In addition, the composition and water temperature of the effluent might be different at another location.

Having said this, our results do indicate that WWTP effluent plumes can cause a behavioural avoidance response in downstream migrating silver eel. When discharge points cannot be avoided, reduced or restricted to areas that are not preferred as safe migration routes, discharge points can be designed such, that the chance is limited that a waterway is (temporarily) impacted over its full width. More attention can be given to determine the dynamics of plumes to establish how often migratory windows occur in time.

## Supporting information

**S1 Appendix. Eel movements 2009.**
(PDF)

**S2 Appendix. Eel movements 2010.**
(PDF)

## Acknowledgments

We would like to thank Dana Allen and Frank Smith from VEMCO for their technical support, Epko Westerhuis for catching the silver eel under difficult circumstances, Peter Paul Schollema from Waterboard *Hunze en Aa's* for his practical support and discussions on the set-up, Eelke de Jong from Waterboard *Noorderzijlvest* for providing information on the WWTP Garmerwolde, and our (former) colleagues Dough Beard and Yannick Friocourt for their assistance in processing the data and the plume modelling, Liesbeth van der Vlies for redrawing the figures. This project was sponsored by the Dutch Ministry of Infrastructure and the Environment via the '*Innovatieprogramma Kaderrichtlijn Water*'.

## Author Contributions

**Conceptualization:** Hendrik Volken Winter, Frank Kleissen, Edwin Matheus Foekema.

**Data curation:** Frank Kleissen, Edwin Matheus Foekema.

**Formal analysis:** Hendrik Volken Winter, Olvin Alior van Keeken, Frank Kleissen, Edwin Matheus Foekema.

**Funding acquisition:** Edwin Matheus Foekema.

**Investigation:** Hendrik Volken Winter, Olvin Alior van Keeken, Frank Kleissen, Edwin Matheus Foekema.

**Methodology:** Hendrik Volken Winter, Olvin Alior van Keeken.

**Project administration:** Hendrik Volken Winter, Edwin Matheus Foekema.

**Resources:** Edwin Matheus Foekema.

**Software:** Frank Kleissen.

**Supervision:** Hendrik Volken Winter.

**Validation:** Frank Kleissen.

**Visualization:** Hendrik Volken Winter, Frank Kleissen, Edwin Matheus Foekema.

**Writing – original draft:** Hendrik Volken Winter, Olvin Alior van Keeken, Frank Kleissen, Edwin Matheus Foekema.

**Writing – review & editing:** Hendrik Volken Winter, Olvin Alior van Keeken, Edwin Matheus Foekema.

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
