## [Decision Letter · Decision Letter 0]

2 May 2023

PONE-D-23-05505Wastewater Plumes Can Act as Non-physical Barriers for Migrating Silver Eel.PLOS ONE

Dear Dr. van Keeken,

Thank you for submitting your manuscript to PLOS ONE. After careful consideration, we feel that it has merit but does not fully meet PLOS ONE’s publication criteria as it currently stands. Therefore, we invite you to submit a revised version of the manuscript that addresses the points raised during the review process.

As you can see, both reviewers were very complementary of your paper and are eager to see the work published but both call for further revision, albeit minor their suggestions will improve the work so please carefully consider all their comments. Please submit your revised manuscript by Jun 16 2023 11:59PM. If you will need more time than this to complete your revisions, please reply to this message or contact the journal office at plosone@plos.org. Please include the following items when submitting your revised manuscript:A rebuttal letter that responds to each point raised by the academic editor and reviewer(s). You should upload this letter as a separate file labeled 'Response to Reviewers'.A marked-up copy of your manuscript that highlights changes made to the original version. You should upload this as a separate file labeled 'Revised Manuscript with Track Changes'.An unmarked version of your revised paper without tracked changes. You should upload this as a separate file labeled 'Manuscript'.If applicable, we recommend that you deposit your laboratory protocols in protocols.io to enhance the reproducibility of your results. Protocols.io assigns your protocol its own identifier (DOI) so that it can be cited independently in the future. For instructions see: https://journals.plos.org/plosone/s/submission-guidelines#loc-laboratory-protocols. Additionally, PLOS ONE offers an option for publishing peer-reviewed Lab Protocol articles, which describe protocols hosted on protocols.io. Read more information on sharing protocols at https://plos.org/protocols?utm_medium=editorial-email&utm_source=authorletters&utm_campaign=protocols.

We look forward to receiving your revised manuscript.

Kind regards,

Dennis M. Higgs

Academic Editor

PLOS ONE

Journal Requirements:

3. Please upload a new copy of Figure 6 as the detail is not clear. Please follow the link for more information: " ext-link-type="uri" xlink:type="simple">https://blogs.plos.org/plos/2019/06/looking-good-tips-for-creating-your-plos-figures-graphics/"
" ext-link-type="uri" xlink:type="simple">https://blogs.plos.org/plos/2019/06/looking-good-tips-for-creating-your-plos-figures-graphics/"

Reviewers' comments:

Reviewer's Responses to Questions

**Comments to the Author**

1. Is the manuscript technically sound, and do the data support the conclusions?

Reviewer #1: Yes

Reviewer #2: Yes

2. Has the statistical analysis been performed appropriately and rigorously? 

Reviewer #1: N/A

Reviewer #2: Yes

3. Have the authors made all data underlying the findings in their manuscript fully available?

Reviewer #1: Yes

Reviewer #2: Yes

4. Is the manuscript presented in an intelligible fashion and written in standard English?

Reviewer #1: Yes

Reviewer #2: Yes

5. Review Comments to the Author

Reviewer #1: This study offers a straightforward objective of interest – is the movement of fish modified by chemical barriers (do fish avoid wastewater plumes in the field). There is ample reason to test this and believe it to be the case, but as the authors note, it has received little attention.

This study is quite small, but well conducted. It adds interesting information and should be published. I have fairly minor comments for consideration. I think one of the figures might need attention and material could be added to the Discussion.

Specific comments

Clear abstract.

40-43. Strictly speaking, the writing is not super strong. ‘been rarely’… ‘may have a chance of…’ but this is a science paper. The message is clear enough. If the authors are after a better product, they can always have it edited.

47. I really like this chemical barrier idea. I don’t think it has been explored in this manner, at least with WW. Nice!

60. To a lot of the world, STP means scientifically treated petroleum. The other way this could go is WWTP effluent, or municipal effluent, but whatever.

91. How much of a difference could it really make? I suggest inserting a number. I bet it is not much and so meaningless in the context of sensor error etc.

113. ‘Eels’ or ‘research subjects’…

194. ‘… are reasonably possible…’ ?

Figure 5. I trust these are real, example traces? If so, please state in the caption. I am puzzled in that you state no attraction was noted, yet you have a trace that indicates at least one animal demonstrated this behavior. Is this behavior example just fiction then? Please clarify. If these are not real traces, please swap out for real ones and indicate no attraction examples existed.

Discussion. I would like to see at least a brief discussion on the variability noted in the observed responses. Is there evidence of behavioral phenotype? This might be interesting. Could something unique be said of the animals that tended to turn away, or perhaps never come back? I do appreciate the dataset is small, but even so, it is worth considering.

Reviewer #2: With much interest I have read the manuscript of Winter and colleagues about an understudied aspect of silver eel migration in Europe. Although non-physical barriers are omnipresent in the Anthropocene, their potential effect on diadromous fish are indeed hardly investigated. As such, this investigation definitely is worthwhile publishing in PlosOne.

I noticed the paper has a certain review history. I agree with the authors that both recent reviewers hardly made any usable recommendations to further improve the paper. As far as I can judge, the paper has however already significantly improved after the first review round.

For me the paper reads fluently, the analysis seem sound, the results are to the point and the discussion focussed to the research questions posed. It is a very concise paper, but according to me this is not necessarily a disadvantage.

I have some minor remarks, primarily about certain aspects that deserve a little bit more explanation.

• Lines 48-50: I doubt that no studies exist on temperature effects of plumes (effluent, cooling water) although the majority might indeed have a focus on oxygen. Maybe good to check and to adapt this statement concordantly.

• Line 101: all transmitters

• Lines 114-115: all eels used in this study were captured downstream the research location. Since all investigated eels were classified as migrating, these eels could potentially already have experienced effects of the STP-effluents during downstream migration. Why were no silver eels from upstream the STP used? Any bias by eel learning behaviour possible here?

• Line 136 a.f.: I can understand the comment by previous reviewers that the methods are a bit too concise for non-specialists to follow. I have some difficulties with the plume modelling. I think it is wise to describe in a small introductory paragraph the outline and build-up of the model. For instance, for me it is not clear to which temperature you refer to in line 156. Does it concern measured temperatures of the effluent and/or the canal water? I assume both temperatures are needed to model the plume dynamics?

• Lines 185-186: What basis do you use for this threshold? Is there literature available of European eels’ or other diadromous fish species’ capacity to sense such or other small temperature differences? Please add references if possible or explain the choice for this threshold.

• Lines 240-243: In my opinion, more explanation is needed to understand this complex figure. At least each subgraph should be mentioned and explained. I see different colours in the track locations, I assume these were related to plume interferences? But in the case of #6, these coloured dots appear far upstream from the modelled flow as shown in the subgraph. By exploring the supplementary materials I noticed a figure that explained the track colours as related to time sequence... This is confusing, please provide a clear explanation directly in the figure legend. The print quality of the figure also seems very poor. I could not read (axis)values in the subgraphs...

• Lines 286-287: I think the effects of non-physical barriers should not be underestimated. You might better point out in the discussion section that migrating eels are subject to a multitude of distracting and attracting flows in the current Anthropocene... maybe one effluent discharge point might have a minor impact but a sequence of many effluent discharge points along the migration route might cause substantial migration delays or extra energy dissipation. In this respect, also the current enrolment of aquathermic applications as renewable energy source within the EU Green deal place another and maybe comparable threat to diadromous fish. You might point this out as well in the discussion.

6. PLOS authors have the option to publish the peer review history of their article (what does this mean?). If published, this will include your full peer review and any attached files.

Reviewer #1: No

Reviewer #2: **Yes: **Jeroen Van Wichelen

---

## [Author Response · Author response to Decision Letter 0]

17 May 2023

Reviewer #1: 

40-43. Strictly speaking, the writing is not super strong. ‘been rarely’… ‘may have a chance of…’ but this is a science paper. The message is clear enough. If the authors are after a better product, they can always have it edited.

- Authors: Changed to: “There is less focus on measures to remove or alter non-physical barriers for fish, like sudden changes in water characteristics. When migrating along waterways near urban areas, fish may have a chance of encountering effluents composed of complex chemical mixtures, that are discharged by sewage treatment plants (WWTPs).”

47. I really like this chemical barrier idea. I don’t think it has been explored in this manner, at least with WW. Nice!

- Authors: no action taken.

60. To a lot of the world, STP means scientifically treated petroleum. The other way this could go is WWTP effluent, or municipal effluent, but whatever.

- Authors: changed into WWTP.

91. How much of a difference could it really make? I suggest inserting a number. I bet it is not much and so meaningless in the context of sensor error etc.

- Authors: According to VEMCO, when they calculated the VPS fish locations, temperature differences between the effluent and the surrounding water had indeed an effect on the precision of the calculated positions. A number however could not be given. 

113. ‘Eels’ or ‘research subjects’…

- Authors: Changed to: Silver eel tagging

194. ‘… are reasonably possible…’ ?

- Authors: We don’t understand this comment

Figure 5. I trust these are real, example traces? If so, please state in the caption. I am puzzled in that you state no attraction was noted, yet you have a trace that indicates at least one animal demonstrated this behavior. Is this behavior example just fiction then? Please clarify. If these are not real traces, please swap out for real ones and indicate no attraction examples existed.

- Authors: Added: fictional. This figure is used to indicate the possible movements silver eel can show. This figure was suggested by another reviewer after our first review cycle, to include each movement. The actual movements of the silver eels in our study are shown in figure 6 and the appendix. Because we want to indicate all possible movements, we keep these fictional ones in this figure.

Discussion. I would like to see at least a brief discussion on the variability noted in the observed responses. Is there evidence of behavioral phenotype? This might be interesting. Could something unique be said of the animals that tended to turn away, or perhaps never come back? I do appreciate the dataset is small, but even so, it is worth considering.

-Authors: Added: Behavioural patterns, however, substantially differed between individuals. This might be related to differences in personality, i.e. behavioural differences among individuals that are consistent through time and across contexts, which were recently found to play a role in passage success of juvenile eels (Mensinger et al. 2021). It might be interesting for future studies to perform personality assays prior to tagging silver eels to address this. 

Reviewer #2: 

• Lines 48-50: I doubt that no studies exist on temperature effects of plumes (effluent, cooling water) although the majority might indeed have a focus on oxygen. Maybe good to check and to adapt this statement concordantly.

- Authors: Changed: our study is the only other study to our knowledge that studies individual behavioural responses to a plume that is not related to oxygen, i.e. hypoxic conditions or super saturation.

Added: Furthermore, studies on the effect of thermal effluents on fish focus mainly on acclimatisation temperature and temperature preferences and not on individual behaviour (e.g. 10-11).

• Line 101: all transmitters

- Authors: changed into “transmitters”

• Lines 114-115: all eels used in this study were captured downstream the research location. Since all investigated eels were classified as migrating, these eels could potentially already have experienced effects of the STP-effluents during downstream migration. Why were no silver eels from upstream the STP used? Any bias by eel learning behaviour possible here?

- Authors: in the canal fishing is not allowed in the upstream area due to the shipping activity. As a result we only were able to obtain eels from an area downstream, caught by a local fishermen who had a fishing license to catch the fish. Added: Fishing is prohibited in the upstream area of the canal.

• Line 136 a.f.: I can understand the comment by previous reviewers that the methods are a bit too concise for non-specialists to follow. I have some difficulties with the plume modelling. I think it is wise to describe in a small introductory paragraph the outline and build-up of the model. For instance, for me it is not clear to which temperature you refer to in line 156. Does it concern measured temperatures of the effluent and/or the canal water? I assume both temperatures are needed to model the plume dynamics?

- Authors: Added: Delft3D-FLOW is a multidimensional (2D or 3D) hydrodynamic (and transport) simulation module within the Delft3D open Source Software (https://oss.deltares.nl/web/delft3d). Delft3D-FLOW calculates non-steady flow and transport phenomena that result from tidal and meteorological forcing on a rectilinear or a curvilinear, boundary fitted grid. The model can include features such as the effect of the earth rotation, space and time varying wind and atmospheric pressures, and simulation of thermal discharge. The main goal of the model was to describe the plume dynamic in time, using the variation in flow rate, flow direction and water temperature in the canal in combination with the flow rate and temperature of the waste water plume.

Added: effluent and the canal water temperature

• Lines 185-186: What basis do you use for this threshold? Is there literature available of European eels’ or other diadromous fish species’ capacity to sense such or other small temperature differences? Please add references if possible or explain the choice for this threshold.

- Authors: Changed into: In the modelling of the plume, temperature was used to identify plume influence, and when temperature increased 0.1 oC from upstream ambient canal temperature, silver eel were considered to be confronted with effluent influence. 

• Lines 240-243: In my opinion, more explanation is needed to understand this complex figure. At least each subgraph should be mentioned and explained. I see different colours in the track locations, I assume these were related to plume interferences? But in the case of #6, these coloured dots appear far upstream from the modelled flow as shown in the subgraph. By exploring the supplementary materials I noticed a figure that explained the track colours as related to time sequence... This is confusing, please provide a clear explanation directly in the figure legend. The print quality of the figure also seems very poor. I could not read (axis)values in the subgraphs...

- Authors: Print quality improved and added: “ Left panel in each of the three examples: visualization of the effluent plume at moment of contact. Middle panel: horizontal movements of the eel within the test area, each dot represents one signal, colour indicates water temperature excess (proxy for contribution of effluent) as experienced by the fish. Right panel: vertical movement of eel (blue line) and modelled excess water temperature (red line).

• Lines 286-287: I think the effects of non-physical barriers should not be underestimated. You might better point out in the discussion section that migrating eels are subject to a multitude of distracting and attracting flows in the current Anthropocene... maybe one effluent discharge point might have a minor impact but a sequence of many effluent discharge points along the migration route might cause substantial migration delays or extra energy dissipation. In this respect, also the current enrolment of aquathermic applications as renewable energy source within the EU Green deal place another and maybe comparable threat to diadromous fish. You might point this out as well in the discussion.

- Authors: added: However migrating silver eels can be subjected to a multitude of distracting and attracting flows, and a sequence of many effluent discharge points along the migration route might can result in substantial migration delays or extra energy dissipation.

---

## [Editor Report · Decision Letter 1]

1 Jun 2023

Wastewater Plumes Can Act as Non-physical Barriers for Migrating Silver Eel.

PONE-D-23-05505R1

Dear Dr. van Keeken,

We’re pleased to inform you that your manuscript has been judged scientifically suitable for publication and will be formally accepted for publication once it meets all outstanding technical requirements.

Kind regards,

Dennis M. Higgs

Academic Editor

PLOS ONE
---

## [Editor Report · Acceptance letter]

14 Jun 2023

PONE-D-23-05505R1 

Wastewater Plumes Can Act as Non-physical Barriers for Migrating Silver Eel. 

Dear Dr. van Keeken:

I'm pleased to inform you that your manuscript has been deemed suitable for publication in PLOS ONE. Congratulations! Your manuscript is now with our production department. 

Kind regards, 

on behalf of

Dr. Dennis M. Higgs 

Academic Editor

PLOS ONE